# European Autism GEnomics Registry (EAGER): protocol for a multicentre cohort study and registry

Madeleine Bloomfield,[1] Alexandra Lautarescu [1,2] Síofra Heraty,[3] Sarah Douglas,[4] Pierre Violland [4] Roderik Plas,[4] Anjuli Ghosh,[4] Katrien Van den Bosch,[4] Eliza Eaton,[5] Michael Absoud,[6,7] Roberta Battini,[8,9] Ana Blázquez Hinojosa,[10] Nadia Bolshakova,[11] Sven Bölte,[12,13,14] Paolo Bonanni,[15] Jacqueline Borg,[16,17,18] Sara Calderoni,[8,9] Rosa Calvo Escalona,[10,19,20,21] Miguel Castelo-Branco [22,23] Josefina Castro-Fornieles,[10,19,20,21] Pilar Caro,[24] Freddy Cliquet,[25] Alberto Danieli,[15] Richard Delorme,[26] Maurizio Elia,[27] Maja Hempel,[24] Claire S Leblond,[25] Nuno Madeira,[23,28,29] Grainne McAlonan,[2,30] Roberta Milone,[8] Ciara J Molloy,[11] Susana Mouga,[23] Virginia Montiel,[31] Ana Pina Rodrigues,[23] Christian P Schaaf,[24] Mercedes Serrano,[31] Kristiina Tammimies,[12,32] Charlotte Tye,[1] Federico Vigevano,[33,34] Guiomar Oliveira,[23,35,36] Beatrice Mazzone,[8] Cara O'Neill,[37] Julie Pender,[38] Verena Romero,[39] Julian Tillmann,[40] Bethany Oakley,[2] Declan G M Murphy,[2,41,42] Louise Gallagher,[11,43,44,45] Thomas Bourgeron,[25] Christopher Chatham,[46] Tony Charman [1]

MB and AL are joint first authors.

For numbered affiliations see end of article.

**Correspondence to**
Dr Alexandra Lautarescu;
alexandra.lautarescu@kcl.ac.uk

## ABSTRACT

**Introduction** Autism is a common neurodevelopmental condition with a complex genetic aetiology that includes contributions from monogenic and polygenic factors. Many autistic people have unmet healthcare needs that could be served by genomics-informed research and clinical trials. The primary aim of the European Autism GEnomics Registry (EAGER) is to establish a registry of participants with a diagnosis of autism or an associated rare genetic condition who have undergone whole-genome sequencing. The registry can facilitate recruitment for future clinical trials and research studies, based on genetic, clinical and phenotypic profiles, as well as participant preferences. The secondary aim of EAGER is to investigate the association between mental and physical health characteristics and participants' genetic profiles.

**Methods and analysis** EAGER is a European multisite cohort study and registry and is part of the AIMS-2-TRIALS consortium. EAGER was developed with input from the AIMS-2-TRIALS Autism Representatives and representatives from the rare genetic conditions community. 1500 participants with a diagnosis of autism or an associated rare genetic condition will be recruited at 13 sites across 8 countries. Participants will be given a blood or saliva sample for whole-genome sequencing and answer a series of online questionnaires. Participants may also consent to the study to access pre-existing clinical data. Participants will be added to the EAGER registry and data will be shared externally through established AIMS-2-TRIALS mechanisms.

**Ethics and dissemination** To date, EAGER has received full ethical approval for 11 out of the 13 sites in the UK (REC 23/SC/0022), Germany (S-375/2023), Portugal (CE-085/2023), Spain (HCB/2023/0038, PIC-164-22), Sweden (Dnr 2023-06737-01), Ireland (230907) and Italy (CET_62/2023, CEL-IRCCS OASI/24-01-2024/ EM01, EM 2024-13/1032 EAGER). Findings will be disseminated via scientific publications and conferences but also beyond to participants and the wider community (eg, the AIMS-2-TRIALS website, stakeholder meetings, newsletters).

## STRENGTHS AND LIMITATIONS OF THIS STUDY

⇒ Data from full genotyping through whole-genome sequencing will be combined with mental and physical health data and participant research priorities.

⇒ The European Autism GEnomics Registry (EAGER) sample (n=1500), although relatively small for genetic analyses, will include a substantial proportion (around one-third) of participants with a rare genetic condition, ensuring that heterogeneous presentations across the autism spectrum are captured.

⇒ The EAGER registry will improve the speed, efficiency and impact of research studies and clinical trials across Europe with a culturally diverse cohort of recontactable participants and shared data.

⇒ EAGER was developed with input from the AIMS-2-TRIALS Autism Representatives and representatives from the rare genetic conditions community.

⇒ Phenotypic data are collected only via self/informant-report questionnaires and not direct clinical assessments.

## INTRODUCTION

Autism is a neurodevelopmental condition diagnosed on the basis of differences in social communication and interaction, sensory processing, restricted and repetitive behaviours, and intense interests,[1 2] with a prevalence of around 1%–2% of the general population.[3] Autism is associated with a very large number of common genetic variants of small effect,[4] as well as with several rare genetic conditions (eg, Dup15q syndrome, Rett syndrome, tuberous sclerosis complex, Angelman syndrome) where autism prevalence is as high as 30%–93%.[5 6]

Many autistic people and people with associated rare genetic conditions have co-occurring mental and physical health conditions that impact quality of life. For example, depression, anxiety, gastrointestinal symptoms and epilepsy commonly co-occur with autism.[7–12] For some autistic people and those with associated rare genetic conditions, many of these conditions can be severe and life-limiting, including respiratory and cardiac disorders in Rett syndrome,[13 14] hyperphagia in Prader-Willi syndrome[15] and tumour growth in tuberous sclerosis complex.[16] This makes the development of treatments and support for these populations a crucial area of research. Further, some autistic people may find aspects of being autistic distressing and disabling and may want help with these.[17–20] All autistic people have the right to evidence-based care and support for aspects that are detrimental to their quality of life and well-being.[21] Despite this, there is currently a paucity of personalised evidence-based care and support for autistic people across the spectrum,[22 23] and polypharmacy (ie, taking multiple medications) for various co-occurring symptoms is commonplace.[24]

With a heritability of around 80%, autism has strong genetic influences.[25 26] However, to date, there is no known gene that, when mutated, increases the likelihood of an autism diagnosis without also increasing the likelihood of intellectual disability or other neurodevelopmental conditions.[27] Consequently, genetic research is an important biomedical tool in the study of autism and associated rare genetic conditions, with benefits including personalised medicine, early identification of health risks, a better self-understanding and a sense of community.[28] To aid the development of evidence-based care for these populations, it is crucial to better understand genetic differences between syndromic forms of autism and autism without a known highly penetrant genetic influence. Further, by linking genetic mechanisms to treatment outcomes, accurate information can be offered regarding potential difficulties, as well as regarding interventions that may be effective for aspects of their lived experience that they want or need support with.

Using extensive genetic and phenotypic data, the European Autism GEnomics Registry (EAGER) will enable a better understanding of the genetic architecture of autism and associated rare genetic conditions, as well as the relationship between genetics and outcomes such as health, quality of life and well-being. The EAGER registry will also allow participants to be recontacted for future research studies or clinical trials that may be suitable for them. This will facilitate research by streamlining and accelerating recruitment across Europe, as well as bring together key research and clinical institutions, to the benefit of participants and their families. In line with tailoring evidence-based care to specific populations, clinical trials that are genetically informed allow researchers to invite participants who are the most suited to, and may benefit most from, a treatment.[29] Selective recruitment at the early stages of treatment development encourages the downstream development of personalised medicine.

This project aims to be a catalyst for future research and clinical trials in autism and associated rare genetic conditions in Europe. This will be achieved by connecting key sites and allowing research groups and pharmaceutical companies planning research studies or clinical trials to have access to a European registry of people who have consented to being recontacted. EAGER data will be shared via established AIMS-2-TRIALS data sharing mechanisms via platforms such as ELIXIR (https://elixir-europe.org/) and the Autism Sharing Initiative (ASI). The ASI is a global collaboration connecting autism data through federated mechanisms.[30] The genetic, phenotypic and clinical data collected in EAGER will be analysed with other international autism datasets such as MSSNG,[31] POND,[32] LEAP,[33] SPARK[34] and Searchlight,[35] providing the opportunity to investigate key research questions.

## METHODS AND ANALYSIS
### Study design and population

EAGER is a European multisite study that aims to create a registry of 1500 adults and children with a diagnosis of autism or a rare genetic condition associated with autism. Recruitment for EAGER will take place at 13 sites across 8 European countries (see figure 1), with the recruitment and data collection protocol being standardised across all sites.

To be eligible, participants need to be over 2 years of age and have a diagnosed rare genetic condition associated with autism and/or have a diagnosis of autism (see figure 2, for examples of conditions). Decisions regarding which conditions and genetic variants to prioritise have been made on the basis of published literature[36] and discussions with experts in the field (TB and LG), with consideration given to their penetrance, prevalence and association with autism. In addition, given the difficulty of recruiting cohorts of meaningful sizes for each of these conditions, pragmatic aspects related to participant access at each of the EAGER sites were also taken into consideration. For participants with a diagnosed rare genetic condition, recruitment will include conditions that are associated with autism or autism-like phenotypes, such as those listed in figure 2. Given that autism assessments are not routinely performed for all people with rare genetic conditions, an autism diagnosis is not required for inclusion into the study for participants in this group. For

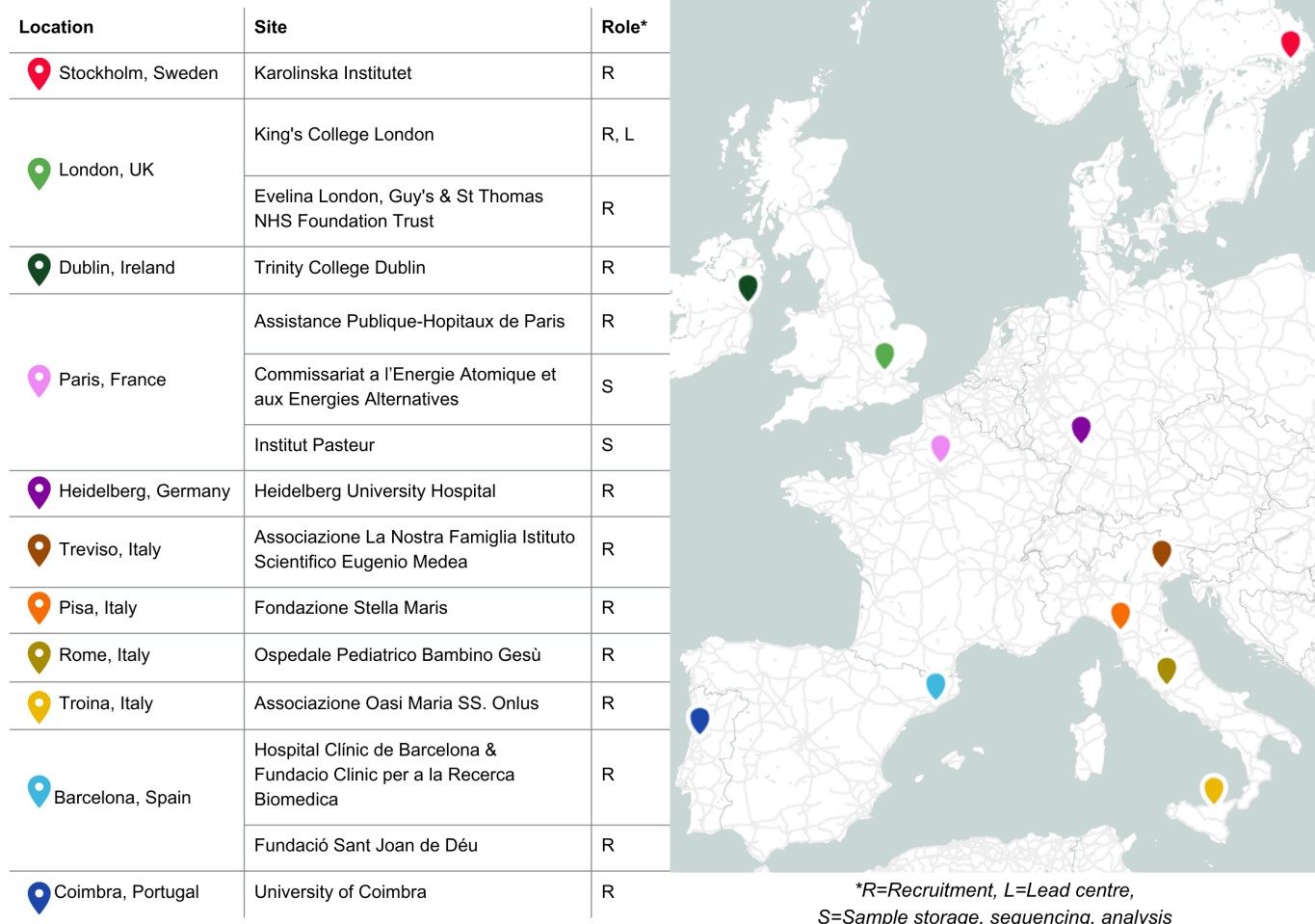

| Location | Site | Role* |
|---|---|---|
| Stockholm, Sweden | Karolinska Institutet | R |
| London, UK | King's College London | R, L |
| | Evelina London, Guy's & St Thomas NHS Foundation Trust | R |
| Dublin, Ireland | Trinity College Dublin | R |
| Paris, France | Assistance Publique-Hopitaux de Paris | R |
| | Commissariat a l'Energie Atomique et aux Energies Alternatives | S |
| | Institut Pasteur | S |
| Heidelberg, Germany | Heidelberg University Hospital | R |
| Treviso, Italy | Associazione La Nostra Famiglia Istituto Scientifico Eugenio Medea | R |
| Pisa, Italy | Fondazione Stella Maris | R |
| Rome, Italy | Ospedale Pediatrico Bambino Gesù | R |
| Troina, Italy | Associazione Oasi Maria SS. Onlus | R |
| Barcelona, Spain | Hospital Clínic de Barcelona & Fundacio Clinic per a la Recerca Biomedica | R |
| | Fundació Sant Joan de Déu | R |
| Coimbra, Portugal | University of Coimbra | R |

*R=Recruitment, L=Lead centre,
S=Sample storage, sequencing, analysis

**Figure 1** EAGER collaborating centres. EAGER, European Autism GEnomics Registry; NHS, National Health Service.

participants with a diagnosis of autism only, and where existing genetic information is available, recruitment will prioritise individuals within multiplex families and individuals with genetic variants associated with autism (see figure 2). Participants will not be excluded from the study based on any other characteristics such as co-occurring conditions, sex or intellectual disability.

### Patient and public involvement and ethical consultations

Genetic research can involve complex ethical challenges, making it crucial to prioritise ethical considerations and participatory research practices.[21 28 37] Participatory research practices can help ensure that the interests of our participants are appropriately represented and that the study is beneficial to its participants and the wider community. EAGER was developed with crucial input and feedback from community representatives via dedicated working groups. Our autism working group consists of AIMS-2-TRIALS Autism Representatives: autistic people and family members or carers of autistic people. Our rare genetic conditions working group consists of parents/carers of people with rare genetic conditions and/or representatives from European rare genetic conditions associations. As EAGER was conceptualised and funded prior to the establishment of our lived experience working groups, it is not

a fully participatory research endeavour. This was discussed with and understood by all members of the working groups at the outset of our collaboration. Within these constraints, every effort was made to consult with our lived experience partners throughout the lifespan of this project, and feedback was integrated to the greatest degree possible.

The EAGER working groups have offered invaluable advice and feedback to guide the planning process of EAGER (see box 1), either as part of working group meetings or through communication via email or contributions to collaborative documents. Open discussion within these groups was encouraged; as the groups included a variety of perspectives, there were naturally different solutions to questions. When no consensus was reached, the core research team made the final decision based on feasibility.

Further, we conducted consultation with other stakeholders, ethical advisors and additional working groups within the consortium, in order to evaluate risks and potentially sensitive implications of this research. We hope that the aforementioned groups will guide EAGER in the most beneficial direction for its participants and the wider community. These groups will continue to be consulted throughout the lifetime of the project.

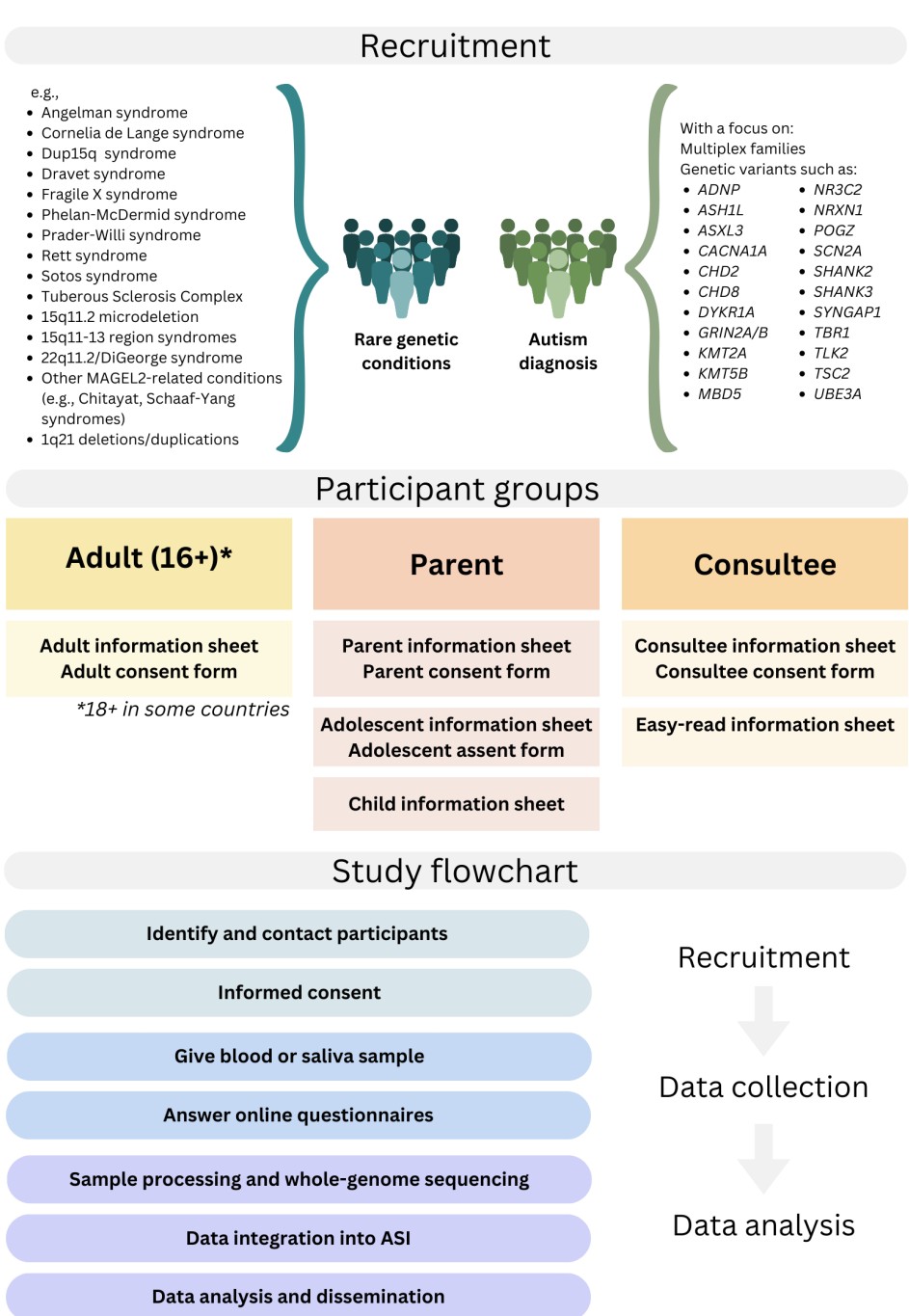

**Figure 2** EAGER protocol summary. ASI, Autism Sharing Initiative; EAGER, European Autism GEnomics Registry.

## Participant recruitment, information sheets and informed consent

Potential participants will be approached by representatives at each local site (ie, members of the clinical or research team). Each site will screen and recruit participants based on recontact of participants from prior research studies or prior/current patients that fit the inclusion criteria. Recruitment targets have been allocated to each of the sites based on local capacity and cohort availability, with targets ranging from n=50 to n=120 per site. The distribution of participant types between sites also differs based on capacity, with some sites recruiting a majority of autistic participants (eg, 90% of target sample) and other sites recruiting a majority of participants with rare genetic conditions (eg, 70% of target sample).

Written informed consent will be required for participation in the study and participants will be provided with information sheets tailored to their age and level of intellectual disability (see figure 2 for details on documents for specific participant groups). A personal consultee will be asked to consent on behalf of individuals who are deemed to not have capacity to consent following a capacity to consent assessment. Due to the high prevalence of co-occurring intellectual disability in people with rare genetic conditions, these may make up a significant

**Box 1    Key examples of how patient and public involvement guided the European Autism GEnomics Registry (EAGER) study**

⇒ Refining the study rationale and clarifying perceived benefit of the study for autistic people.
⇒ Deciding on study name and logo.
⇒ Discussing ethical implications of the research, ensuring that appropriate safeguards are in place and that any limitations are transparently communicated to participants.
⇒ Ensuring that the information presented in study documents is clear, accessible, transparent and sufficient to allow participants to make an informed decision about their participation.
⇒ Creating a 'Data sharing Frequently Asked Questions' document to accompany the information sheet.
⇒ Developing different versions of information sheets tailored to the needs of our different participant groups (eg, easy-read, adolescent information sheet).
⇒ Ensuring that the consent forms are clear and transparent about what precisely a participant is consenting to.
⇒ Ensuring that the language used in study documents is appropriate and in line with community preferences.
⇒ Discussing the phenotypic measures used in the study and suggesting changes to the content/language of questionnaires.
⇒ Sharing all key study documents for feedback prior to submission for ethical approval.
⇒ Clearly stating that EAGER is against eugenics/prenatal testing for autism on participant-facing documents.
⇒ Detailing exactly what analyses will be performed on participant data and making it clear that no further analyses will occur without reconsenting.
⇒ Clarifying that analyses performed on EAGER data will need to be approved by a committee before they can begin.
⇒ Clarifying in the ethics documents that not all decisions were made based on consensus.
⇒ Adding a disclaimer in the protocol/ethics with regard to the fact that citing papers does not represent an endorsement of the language/aims of those papers or of the authors.

proportion of the overall sample. Parental consent will be required for those under the legal age of consent (eg, 16 years old in the UK).

To take part in EAGER, participants must consent to core aspects of the study: giving a blood or saliva sample (or permission to use a pre-existing sample), completing questionnaires, being added to the registry, and data sharing.

### Biological sample collection
Participants will give a blood or saliva sample, depending on preference and appropriateness. For blood samples, peripheral venous blood will be collected in EDTA-coated vacucontainers at each local site by experienced phlebotomists or other professionals who are trained in collecting blood samples. For saliva collection, the Oragene•DNA self-collection kits, as well as assisted collection kits for those unable to collect their own sample, will be used. If existing blood or saliva samples are available and have been stored appropriately, they can be used in lieu of a new sample, with participant consent.

### Genetic analysis
#### DNA quality control
All concentrations will be measured in fluorescence (Quant-IT kit), in duplicate; samples with low concentrations will be quantified with a more sensitive kit. Samples with initial concentrations higher than 120 ng/µL will be diluted to concentrations compatible with the working conditions of the platform and samples with heterogeneous results in the first measurements will be verified by a second series of measurements. The quality of the samples will be evaluated on all samples by migrating a small amount of DNA on TapeStation Agilent 4200, as well as a PCR amplification test (simultaneous amplification of two microsatellite markers) and a verification of the sex of individuals by PCR. With few exceptions, samples with a DNA Integrity Number <6 or showing a sex mismatch with the file information will be excluded.

#### Whole genome sequencing
After quality control, the preparation of the libraries will be performed on the automated platform of the National Center for Genomic Research (CNRGH). Fragmentation of the genomic DNA with ultrasound (Covaris) will use the protocol of the lllumina TruSeq DNA PCR-free kit with a starting quantity of 1 µg of genomic DNA. Quality control will be systematically performed after preparation of the libraries by electropherogram verification (Bioanalyzer 3 profile) and quantitative PCR (Light cycler 480, Roche). Clustering capacity will be assessed by sequencing 1 million reads on Miseq. Sequencing will be performed in pair-end (2×150 bases) on the Illumina NovaSeq X platform of the CNRGH, using the corresponding sequencing kits and according to the supplier's specifications. Quality control will be performed during sequencing (intensity, Q30, phasing/prephasing/base proportion). A minimum quantity of sequences allowing an average depth of about 30x for each sample will be generated.

#### Variant calling single-nucleotide variants, indels and CNVs
After aligning the genomes to the GRCh38 human reference, variants will be called jointly for the cohort using DeepVariant V.1.6.0 and GLnexus V.1.4.1. All multiallelic variants will be decomposed and indels will be normalised using bcftools 1.15 with the norm mode. Functional annotation will be performed using VEP Ensembl 111. Copy number variants (CNVs) will be identified using ClinSV V.1.0.1, which combines Lumpy and CNVnator to look for discordantly mapping read pairs, split-mapping reads and depth of coverage changes. A table with rare gene affecting variants will be generated containing all variants passing the variant detection threshold (PASS and HIGH) and having a population allele frequency≤1%.

#### Variant validation
Validation on Compressed Reference-oriented Alignment Maps will use Integrative Genomics Viewer (IGV): for single-nucleotide variants and Indels, after building a

list of candidate de novo variants using slivar 0.2.8, we will exclude those located in Low Complexity Regions. We will then use a machine learning approach to predict for each of those candidates de novo variants whether they were true or false positives. This approach will be calibrated using a learning dataset of ~900 variants manually validated from independent cohorts sequenced using the same methodology, allowing a false positive rate of ≤5%. We will then validate the CRAM files visually using IGV for all the de novo variants in genes related to neurodevelopmental conditions. Sanger sequencing will be used to test de novo loss-of-function or missense with Missense badness, PolyPhen-2, and Constraint scores ≥2.

### Ancestry

Principal components analysis of the variance standardised relationship matrix will be used to evaluate the ancestry of individuals in the EAGER cohort and to provide components for any covariate adjustments. To find subpopulation clusters, we will compute the admixture (admixture 1.3.0) of the cohort along with a reference population (eg, 1000 genomes, Simons Genome Diversity Project), and then apply a machine learning approach to predict ancestries.

### Polygenic scores

Each participant will have a polygenic profile on common variants computed from WGS. Polygenic scores will be computed using state-of-the-art approaches such as SBayesR (GCTB 1.05beta) or SBayesRC 0.2.4. Each polygenic score will be standardised on neurotypical individuals according to their ancestry.

### Genetic profile of participants

Participants will have a genetic profile that includes rare and common variants associated with neurodevelopmental conditions. Several databases (SFARI/SPARK, DBD, SysNDD, DDG2P) have been developed to provide extensive information on these genes. An updated list of more than 1800 genes associated with neurodevelopmental conditions (ie, high-confidence genes) can be downloaded on the Gene Trek website. These are genes with strong evidence for genetic association with neurodevelopmental conditions (ie, SFARI 'category 1' genes, DBD 'tier 1' or 'autosomal recessive' inheritance list, DDG2P 'definitive' confidence category and 'brain/cognition' organ specificity, SysNDD 'definitive' category and all SPARK genes). For common variants through the polygenic scores, a list of summary statistics for traits in different domains of interest will be built (eg, psychiatric traits such as anxiety, neurodevelopmental conditions such as autism or personal traits such as subjective well-being).

### Measures

Following collection of the biological sample, participants will be given access to an online form hosted on the Qualtrics platform, with a series of questionnaires, focusing on autism, rare genetic conditions, co-occurring physical and mental health conditions, opinions on research priorities, and quality of life and well-being (see table 1). These will be a mixture of validated and non-validated questionnaires that have been adapted for EAGER and have been chosen to overlap with other major initiatives or studies both within AIMS-2-TRIALS[38 39] and externally, such as MSSNG,[31] POND,[32] LEAP,[33] SPARK[34] and Simon's Searchlight.[35] All questionnaires have been translated into the local language of each site from English. The online questionnaires will be completed by participants answering on behalf of themselves, or by parents and consultees answering on behalf of the participant.

### Data analysis

EAGER will collect genetic and phenotypic data, which can be analysed both alone and in conjunction with other data collected within AIMS-2-TRIALS, as well as with shared data from other cohorts (eg, via ELIXIR or the ASI). Research questions will be decided in collaboration with our community working groups, but a few brief examples are outlined below.

Using the genetic and phenotypic data, we can answer questions such as:
► Identifying genetic variants that are associated with an autism diagnosis and/or autism phenotypic characteristics (eg, scores on Social Responsiveness Scale and Social Communication Questionnaire).
► Identifying genetic variants that are associated with co-occurring conditions such as neurodevelopmental (eg, scores on Conners' Adult ADHD Rating Scales and ADHD rating scale) or mental health conditions (eg, scores on Strengths and Difficulties Questionnaire, Patient Health Questionnaire-9, and Generalised Anxiety Disorder-7).
► Identifying genetic variants that are associated with characteristics such as quality of life (eg, Cantrill ladder, WHO Quality of Life-BREF (WHO-QOL-BREF)) or functioning (eg, general functioning questionnaire).

Using the phenotypic data, we can answer questions such as:
► Identifying participants' priorities for future research (eg, Voice of the Community survey) and assessing the factors that may influence responses (eg, diagnoses, age).
► Identifying associations between participant diagnoses (including co-occurring conditions) and subjective experiences such as quality of life (eg, Cantrill ladder, WHO-QOL-BREF) or mental well-being (eg, Warwick-Edinburgh Mental Well-being Scales).

The sample size of this study (n=1500), which was driven by the available budget, is moderate-to-large in the field and will be significantly enhanced via the planned data sharing and integration. While EAGER will not recruit a control group, we expect there to be significant opportunities for analyses where participant subgroups can be identified and compared (eg, participants with epilepsy compared with participants without epilepsy). The deep

**Table 1** EAGER questionnaires

| Domain | Measure | Description | Respondents |
|---|---|---|---|
| Autism traits | Social Responsiveness Scale[43] | 65 items: Autism traits (including social awareness, cognition, communication, motivation, and repetitive interests and behaviours) | Adult self-report |
| | Social Communication Questionnaire[44] | 40 items: Autism traits (including communication skills and social functioning) | Parent or consultee report for child or adult |
| Mental health | Strengths and Difficulties Questionnaire[45] | 25 items: General mental health across five scales (conduct problems, hyperactivity, emotional problems, peer problems, prosocial behaviour) | Parent or consultee report for adult or child |
| | Patient Health Questionnaire-9 (PHQ-9)[46] | 9 items: Core features of major depressive disorder | Adult self-report |
| | PHQ-A[47] | 9 items: Core features of major depressive disorder* | Adolescent (16 and 17 years) self-report |
| | Generalised Anxiety Disorder-7[48] | 7 items: Core features of generalised anxiety disorder | Adult self-report |
| Quality of life | WHO Quality of Life-BRE[49] | 26 items: Quality of life across four domains (physical health, psychological health, social relationships, environment) | Adult self-report |
| | Cantrill Ladder[50] | 1 item: Subjective quality of life (10 point scale from 'worst possible life' to 'best possible life') | Adult self-report. Parent or consultee report for child or adult. |
| Mental well-being | Warwick-Edinburgh Mental Well-being Scales[51] | 14 items: Mental well-being and health | Adult self-report. Parent or consultee self-report† |
| Restrictive and repetitive behaviours | Adult Routines inventory[52] | 55 items: Restrictive and repetitive behaviours | Adult self-report |
| | Repetitive Behaviours Scale-Revised[53] | 43: Assessing restrictive and repetitive behaviours | Parent or consultee report for child or adult |
| Attention Deficit Hyperactivity Disorder (ADHD) | Conners' Adult ADHD Rating Scales[54] | 26 items: Core features of ADHD | Adult self-report |
| | ADHD Rating Scale[55] | 18 items: Core features of ADHD | Parent report for child Consultee report for adult‡ |
| Demographics | Non-validated questionnaire | 16–25 items: Demographic characteristics | Adult self-report. Parent or consultee report for child or adult. |
| Co-occurring conditions | Non-validated questionnaire (adapted from Simons Searchlight[35]) | Items assessing 168 diagnosed and self-diagnosed psychiatric, neurodevelopmental, and physical health conditions. | Adult self-report. Parent or consultee report for child or adult. |
| Medications | Non-validated questionnaire | 1 item: Number of medications taken by participant per day | Adult self-report. Parent or consultee report for child or adult. |
| General functioning | Non-validated questionnaire (adapted from[56]) | 7–11 items: Seizures, gross motor skills, language and communication, intellectual disability, support needs, sleep | Adult self-report. Parent or consultee report for child or adult. |
| Voice of the community | Non-validated questionnaire (adapted from Simon Searchlight[35]) | 7–13 items: Priorities for research | Adult self-report. Parent or consultee report for child or adult. |

*Additional questions and assessment of suicidality in the original measure were removed for the purpose of this study and only the first core 9 items were included.
†Administered to parents and caregivers in relation to their own mental well-being.
‡The ADHD rating scale was adapted for use in EAGER to make it appropriate for adults and remove child-specific language.
EAGER, European Autism GEnomics Registry.

phenotyping and the expected heterogeneity within the sample will enable us to better understand differences between autistic people with and without a known genetic cause.

## Data sharing and eager registry
### Data sharing
Currently, funding bodies and participant groups have identified the need for research groups to share their

data to maximise the value of each research contribution, pool data to address research questions that require a larger number of participants or to carry out meta-analyses.[40] For this reason, EAGER data will be shared with the autism research community. Data from AIMS-2-TRIALS studies (including EAGER) are stored centrally on the OWEY platform developed by and hosted at Institut Pasteur in Paris, France. This enables researchers within the AIMS-2-TRIALS consortium to access and use the data. External data sharing is done via the ELIXIR platform, in accordance with the AIMS-2-TRIALS data sharing policy. In addition, EAGER data will be shared via the ASI platform, as detailed below. With participant consent, EAGER data (genetic data, data from online questionnaires and existing data held at local sites) will be shared with any explicit identifiers removed. Rigorous safeguards will be implemented for access to data via established AIMS-2-TRIALS mechanisms and via the ASI platform.[30] Regardless of the data sharing mechanism, AIMS-2-TRIALS researchers will act as data custodians and retain control over who the data are shared with and accessed by. Researchers will not be able to access EAGER data without approval from a data access committee, which will consist of autistic and non-autistic people, researchers and members of the rare genetic community.

### Autism Sharing Initiative

The ASI is a new collaboration working to create the first federated, global network for sharing genomic and clinical data in autism research. The ASI is led by software company DNAStack and involves other collaborators such as Autism Speaks, Roche and the Ontario Brain Institute (please see[41] for the full list). This is an ongoing project and new collaborators may be added.

Data sharing within the ASI will be built on decentralised, federated data sharing principles. Data federation allows researchers to search, access, and analyse data that reside in different locations and are connected via a network, without the need for data to be moved between institutions/researchers. Data federation allows researchers to interrogate and analyse data hosted in different locations, in compliance with data regulations and improving privacy and security. A federated data network involves connecting individual datasets, which remain in their protected local environments (ie, at the research institution where the data were collected), across one cloud-based network. To our knowledge, the ASI carries no additional risk to participant reidentification than traditional data sharing. The ASI software is built on security standards developed by the Global Alliance for Genomics and Health[42] and regularly updated in order to ensure that data are safeguarded. Within the ASI, data will include genomic, multiomic, medical and clinical data from international autism research sources.

### EAGER registry

The EAGER registry will allow participants to be contacted for future clinical trials and research studies. Researchers viewing shared participant data (ie, regardless of data sharing mechanism) will be able to contact the coordinating site (King's College London) to request specific participant groups based on their desired criteria without access to participant's personal information. King's College London will pass these requests on to the relevant local sites, which can send information about the study or clinical trial to participants. AIMS-2-TRIALS is funded until May 2025. Sustainability is a core aspect of the work done as part of the consortium and plans are in place to ensure that EAGER will continue to be supported through future funding initiatives currently being pursued by the investigators.

## ETHICS AND DISSEMINATION
### Ethics

EAGER has received full ethical approval for 11 out of the 13 sites, with approval pending from an ethics committee in France and one in Italy. Approval has been received for all sites in the UK (REC 23/SC/0022), Germany (S-375/2023), Portugal (CE-085/2023), Spain (HCB/2023/0038, PIC-164–22), Sweden (Dnr 2023-06737-01) and Ireland (230907) and three sites in Italy (CET_62/2023, CEL-IRCCS OASI/24-01-2024/EM01, EM 2024-13/1032 EAGER). The protocol described here is being followed by each of the sites and is based on the approved UK documents. Please refer to the above section 'Patient and Public Involvement and Ethics Consultation' for details on ethics and discussions within our community working groups.

### Dissemination

We plan to disseminate our findings widely via academic avenues, such as publications in peer-reviewed major international scientific journals and conferences. In addition, we will disseminate beyond academic avenues to ensure that our findings reach participants and the wider communities, including the AIMS-2-TRIALS website (www.aims-2-trials.eu), stakeholder meetings, and newsletters. As part of the overall AIMS-2-TRIALS project, we have access to a European network including researchers, clinicians, and families within the autism community. This network will also provide a forum for the dissemination of research findings and training opportunities.

**Author affiliations**
[1]Department of Psychology, Institute of Psychiatry, Psychology & Neuroscience, King's College London, London, UK
[2]Department of Forensic and Neurodevelopmental Sciences, Institute of Psychiatry, Psychology, & Neuroscience, King's College London, London, UK
[3]Department of Psychological Sciences, Birkbeck University of London, London, UK
[4]AIMS-2-TRIALS A-Reps, Cambridge University, Cambridge, UK
[5]Autism Research Centre, Cambridge University, Cambridge, UK
[6]Department of Children's Neurosciences, Evelina London Children's Hospital, Guy's and St Thomas' Hospitals NHS Trust, London, UK
[7]Department of Women and Children's Health, Faculty of Life Sciences and Medicine, School of Life Course Sciences, King's College London, London, UK

[8]Department of Developmental Neuroscience, IRCCS Fondazione Stella Maris, Pisa, Italy

[9]Department of Clinical and Experimental Medicine, University of Pisa, Pisa, Italy

[10]Department of Child and Adolescent Psychiatry and Psychology, Institute of Neurosciences, Hospital Clinic Universitari Barcelona, Barcelona, Spain

[11]Department of Psychiatry, School of Medicine, Trinity College Dublin, Dublin, Ireland

[12]Center of Neurodevelopmental Disorders (KIND), Department of Women's and Children's Health, Centre for Psychiatry Research, Karolinska Institutet, Stockholm, Sweden

[13]Child and Adolescent Psychiatry, Stockholm Health Care Services, Stockholm, Sweden

[14]Curtin Autism Research Group, Curtin School of Allied Health, Curtin University, Perth, Western Australia, Australia

[15]Epilepsy Unit, Scientific Institute IRCCS E. Medea Conegliano, Treviso, Italy

[16]Centre for Psychiatry Research and Centre for Cognitive and Computational Neuropsychiatry (CCNP), Department of Clinical Neuroscience, Karolinska Institutet & Stockholm Health Care Services, Stockholm, Sweden

[17]Department of Neuropsychiatry, Region Västra Götaland, Sahlgrenska University Hospital, Gothenburg, Sweden

[18]Department of Psychiatry and Neurochemistry, Institute of Neuroscience and Physiology, Sahlgrenska Academy at The University of Gothenburg, Gothenburg, Sweden

[19]Institut d'Investigacions Biomèdiques August Pi i Sunyer, Barcelona, Spain

[20]Centro de Investigación Biomédica en Red de Salud Mental (CIBERSAM), Madrid, Spain

[21]Department of Medicine, Institute of Neuroscience, University of Barcelona, Barcelona, Spain

[22]Institute of Physiology, Faculty of Medicine, University of Coimbra, Coimbra, Portugal

[23]Coimbra Institute for Biomedical Imaging and Translational Research (CIBIT), Institute of Nuclear Sciences Applied to Health (ICNAS), University of Coimbra, Coimbra, Portugal

[24]Institute of Human Genetics, University Hospital Heidelberg, Heidelberg, Germany

[25]Génétique Humaine et Fonctions Cognitives, UMR3571 CNRS, Institut Pasteur, Paris, France

[26]Child and Adolescent Psychiatry Department, Robert Debre Hospital, APHP, Paris, France

[27]Unit of Neurology and Clinical Neurophysiopathology, Oasi Research Institute-IRCCS, Troina, Italy

[28]Psychiatry Department, Centro Hospitalar e Universitário de Coimbra EPE, Coimbra, Portugal

[29]Institute of Psychological Medicine, Faculty of Medicine, University of Coimbra, Coimbra, Portugal

[30]Behavioural and Developmental Clinical Academic Group, South London and Maudsley NHS Foundation Trust, London, UK

[31]Pediatric Neurology Department, Hospital Sant Joan de Déu, Institut de Recerca Sant Joan de Deu, Barcelona, Spain

[32]Astrid Lindgren Children's Hospital, Karolinska University Hospital, Stockholm, Sweden

[33]Neurological Sciences and Rehabilitation Medicine Scientific Area, Bambino Gesù Children's Hospital, Rome, Italy

[34]Paediatric Neurorehabilitation Department, IRCCS San Raffaele, Rome, UK

[35]University Clinic of Pediatrics, Faculty of Medicine, University of Coimbra, Coimbra, Portugal

[36]Child Developmental Center and Research and Clinical Training Center, Pediatric Hospital, Centro Hospitalar e Universitário de Coimbra (CHUC), Coimbra, Portugal

[37]Cure Sanfilippo Foundation, Columbia, South Carolina, USA

[38]SYNGAP Research Fund, San Diego, California, USA

[39]Dup15q e.V, Hessen, Germany

[40]Roche Pharma Research and Early Development, Roche Innovation Center, Basel, Switzerland

[41]South London and Maudsley NHS Foundation Trust, London, UK

[42]Institute for Translational Neurodevelopment, Institute of Psychiatry, Psychology & Neuroscience, King's College London, London, UK

[43]SickKids Research Institute, Peter Gilgan Centre for Research and Learning, The Hospital for Sick Children, Toronto, Ontario, Canada

[44]Child and Youth Division Centre for Addiction and Mental Health, CAMH, Toronto, Ontario, Canada

[45]Department of Psychiatry, Temerty Faculty of Medicine, Univeristy of Toronto, Toronto, Ontario, Canada

[46]F Hoffmann-La Roche Ltd, Basel, Switzerland

**Contributors** Conceptualisation: MB, AL, SH, SD, PV, RP, AG, KVdB, EE, CO'N, VR, JT, BO, DGMM, LG, TB, CC and TC. Funding acquisition: DGMM, CC and TC. Project administration: MB, AL, DGMM, LG, TB and CC. TC. Supervision: AL and TC. Visualisation: MB and AL. Data collection: MB, AL, MA, RB, ABH, NB, SB, PB, JB, SC, RCE, MC-B, JC-F, PC, AD, RD, ME, MH, NM, GM, RM, CJM, SM, VM, APR, CPS, MS, KT, CT, FV, GO, BM, LG and TC. Writing-original draft: MB, AL and TC. Writing-review and editing: all authors.

**Funding** This work has received funding from the Innovative Medicines Initiative 2 Joint Undertaking under grant agreement No 777394 for the project AIMS-2-TRIALS. This Joint Undertaking receives support from the European Union's Horizon 2020 research and innovation programme and EFPIA and AUTISM SPEAKS, Autistica, SFARI and from Horizon Europe (grant agreement no. 101057385) and from UK Research and Innovation (UKRI) under the UK government's Horizon Europe funding guarantee (grant no.10039383) (R2D2-MH). In addition, SB receives funding from Swedish Research Council and Region Stockholm. CT receives funding from Epilepsy Research UK, Autistica, the Baily Thomas Charitable Fund and the Tuberous Sclerosis Association. LG receives funding from Horizon Europe: Risk and Resilience in Developmental Diversity and Mental Health (R2D2-MH). CPS receives funding from Foundation of Prader-Willi Research, European Joint Program for Rare Diseases, Illumina, German Children's Cancer Foundation and the German Ministry of Education and Research. RB, RM, SC, BM receive funding from the Italian Ministry of Health (Ricerca Corrente and 5x1000 to IRCCS Fondazione Stella Maris). The EAGER study is funded by a Direct Financial Contribution by F Hoffmann La-Roche Ltd as part of the AIMS-2-TRIALS consortium funded by IMI/IHI. This paper represents independent research part-funded by the NIHR Maudsley Biomedical Research Centre at South London and Maudsley NHS Foundation Trust and King's College London.

**Disclaimer** The funders had no role in the design of the study; in the collection, analyses, or interpretation of data; in the writing of the manuscript, or in the decision to publish the results. Any views expressed are those of the author(s) and not necessarily those of the funders.

**Competing interests** In the past 3 years, TC has served as a paid consultant to F. Hoffmann-La Roche and Servier and has received royalties from Sage Publications and Guilford Publications. DGMM has received funding for a PhD studentship from Compass, and for consulting from Jaguar Therapeutics and Hoffman Le Roche. GM receives funding for an investigator-initiated study from Compass Pathways; no financial or other conflict of interest with the present study. SB discloses that he has in the last 3 years acted as an author, consultant, or lecturer for Medice, Roche and Linus Biotechnology. SB receives royalties for textbooks and diagnostic tools from Hogrefe, UTB, Ernst Reinhardt, Kohlhammer, and Liber, and is a partner at NeuroSupportSolutions International AB. CC is a full-time employee of Genentech and owns stocks or RSUs in Roche Holdings. MA is the UK chief investigator for a trial sponsored by Roche (a phase II, randomised, double-blind, placebo-controlled, parallel group study to evaluate the safety, efficacy and pharmacodynamics of 52 weeks of treatment with basmasanil in participants aged 2–14 years old with dup15q syndrome followed by a 2-year optional open-label extension). LB served on an advisory board to Kingdom therapeutics in 2022.

**Patient and public involvement** Patients and/or the public were involved in the design, or conduct, or reporting, or dissemination plans of this research. Refer to the Methods section for further details.

**Patient consent for publication** Not applicable.

**Provenance and peer review** Not commissioned; externally peer reviewed.

**ORCID iDs**
Alexandra Lautarescu http://orcid.org/0000-0001-7680-8936
Pierre Violland http://orcid.org/0000-0003-2346-8039
Miguel Castelo-Branco http://orcid.org/0000-0003-4364-6373
Tony Charman http://orcid.org/0000-0003-1993-6549

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
