## [Reviewer comments · BMJ Open]

ARTICLE DETAILS

TITLE (PROVISIONAL)	European Autism GENomics Registry (EAGER): Protocol for a multicentre cohort study and registry
AUTHORS	Bloomfield, Madeleine; Lautarescu, Alexandra; Heraty, Síofra; Douglas, Sarah; Violland, Pierre; Plas, Roderik; Ghosh, Anjali; Van den Bosch, Katrien; Eaton, Eliza; Absound, Michael; Battini, Roberta; Blázquez Hinojosa, Ana; Bolshakova, Nadia; Bölte, Sven; Bonanni, Paolo; Borg, Jacqueline; Calderoni, Sara; Calvo Escalona, Rosa; Castelo-Branco, Miguel Editorial Board Member; Castro-Fornieles, Josefina; Caro, Pilar; Cliquet, Freddy; Danieli, Alberto; Delorme, Richard; Elia, Maurizio; Hempel, Maja; Leblond, Claire; Madeira, Nuno; McAlonan, Grainne; Milone, Roberta; Molloy, Ciara J.; Mougá, Susana; Montiel, Virginia; Pina Rodrigues, Ana; Schaaf, Christian P.; Serrano, Mercedes; Tammimies, Kristiina; Tye, Charlotte; Vigevano, Federico; Oliveira, Guiomar; Mazzone, Beatrice; O'Neill, Cara; Pender, Julie; Romero, Verena; Tillmann, Julian; Oakley, Bethany; Murphy, Declan G. M.; Gallagher, Louise; Bourgeron, Thomas; Chatham, Christopher; Charman, Tony

VERSION 1 – REVIEW

REVIEWER	Moassefi, Mana Mayo Clinic
REVIEW RETURNED	20-Nov-2023

GENERAL COMMENTS	This protocol manuscript is explaining clearly about the process of genomic data recruitment for Autistic patients. The rationale and methodology is clearly explained. Just a few quick comments: 1- The introduction is not very well written. For example lines 196-200 there is a very long sentence, and also not very academically written. lines 202 to 209 is a part of methodology which saying it with this level of detail in this section is unnecessary. 2- I think authors can elaborate more on why they only picked these rare genetic syndromes. Do they have a threshold for autism prevalence among genetic conditions that only these syndromes were above this threshold or is there any other reason?
---

REVIEWER	Yip, Benjamin Hon Kei The Chinese University of Hong Kong, School of Public Health and Primary Care
REVIEW RETURNED	13-Mar-2024

GENERAL COMMENTS	The EAGER study is timely and has the potential to be a successful platform for facilitating recruitment for future trials and research studies. I have a few minor comments that I hope will enhance the clarity of this manuscript: 1. **Aims and Objectives** : The primary aim is clearly articulated
---

	and pertinent to the field of autism and rare genetic conditions. However, the methodology for achieving the secondary aim is unclear, particularly since participants in EAGER are either autistic or have rare genetic conditions related to autism. Without a proper control group, it will be challenging to study the association between mental and physical health traits and participants' genetic profiles. While studying the association of a continuous trait, such as the Social Responsiveness Scale (SRS), with the genetic profile may be feasible, this hinges on whether the ascertained sample provides sufficient heterogeneity. Additionally, the manuscript does not clarify whether the sample size possesses adequate statistical power to support such an association study, nor does it address the quota allocation for ASD and non-ASD cases (but with related rare genetic conditions). 2. Methodology: Additional details regarding recruitment strategies, data collection methods, and plans for data analysis would be beneficial. For instance, the rationale behind the target sample size of N=1,500 is not specified. Is this number a result of budget constraints, subject recruitment capabilities, or a power calculation for the secondary aim? How will the 1,500 cases be allocated across different sites? Is there a quota sampling strategy in place (e.g., 50% ASD, 50% rare-genes)? Furthermore, the manuscript lacks a description of the genetic data sequencing process, such as the standardized platform and arrays used. Highlighting EAGER's standardized procedures, which I believe is one of its strengths, is recommended. 3. Ethical Considerations: The ethical considerations, including informed consent, confidentiality, data protection, and participant privacy safeguards, are thoroughly addressed, demonstrating strong stakeholder engagement. 4. Data Management: It is unclear whether ASI will serve as the host for the federated system of EAGER. The AIMS-2 Trial's website mentions the collection of data and the creation of a database that is accessible, practical, and easy to navigate. Does this refer to EAGER? Will the AIMS-2 trial data be managed by ASI (DNAstack)? Since the AIMS-2 trial was initiated a few years ago, utilizing the already collected data for the development of EAGER could be more cost-effective. 5. Feasibility and Sustainability: The EAGER study is funded by grant agreement No. 777394 for the AIMS-2 trials project. Does this imply that EAGER is the database referenced on the AIMS-2 trial website? If so, leveraging the existing AIMS-2 trial data for the establishment of EAGER would seem prudent. If that is the plan, then additional details about AIMS-2, particularly its timeline, are necessary. Another concern pertains to sustainability: is there sufficient funding or a realistic plan in place to ensure the registry's longevity (e.g., management fees to ASI)? One minor comments: In Figure 1. What does R, L, S mean?
--	---

VERSION 1 – AUTHOR RESPONSE

Reviewer 1 Dr. Mana Moassefi, Mayo Clinic

This protocol manuscript is explaining clearly about the process of genomic data recruitment for Autistic patients. The rationale and methodology is clearly explained.

Reviewer 1 Comment 1

1- The introduction is not very well written. For example lines 196-200 there is a very long sentence, and also not very academically written. lines 202 to 209 is a part of methodology which saying it with this level of detail in this section is unnecessary.

In response to this comment we have made some changes to the introduction, with the view of increasing clarity. We have shortened and clarified the sentence that was in lines 196-200 in the original version of the manuscript, and moved some information to the methods section, as advised by the reviewer.

Reviewer 1 Comment 2

2- I think authors can elaborate more on why they only picked these rare genetic syndromes. Do they have a threshold for autism prevalence among genetic conditions that only these syndromes were above this threshold or is there any other reason?

We thank the reviewer for this comment. The list of rare genetic syndromes is not an exclusive list, but a list of examples. We have now clarified this both in text (see bullet point below), as well as in figure 2.

- “To be eligible, participants need to be over 2 years of age and have a diagnosed rare genetic condition associated with autism and/or have a diagnosis of autism (see figure 2 for examples of conditions). Decisions regarding which conditions and genetic variants to prioritise have been made on the basis of published literature (36) and discussions with experts in the field (TB, LG), with consideration given to the penetrance, prevalence, and association with autism. In addition, given the difficulty of recruiting cohorts of meaningful sizes for each of these conditions, pragmatic aspects related to participant access at each of the EAGER sites were also taken into consideration. For participants with a diagnosed rare genetic condition, recruitment will include conditions that are associated with autism or autism-like phenotypes, such as those listed in figure 2. Given that autism assessments are not routinely performed for all people with rare genetic conditions, an autism diagnosis is not required for inclusion into the study for participants in this group.”

We did not include a specific threshold for autism prevalence given the discrepancies in the literature with regards to prevalence within each disorder (e.g., range of 15-50% for 22q11.2 deletion syndrome), as well as inconsistencies in diagnosis (i.e., not everyone with a rare genetic condition will need or have access to an autism assessment). However, all of the listed syndromes have a much higher prevalence of autism relative to the general population (i.e., 1-2%). The list of conditions illustrated in Figure 2 is not exhaustive, and other rare genetic conditions may be included on a case by case basis following discussion within the research team.

Reviewer 2 Prof. Benjamin Hon Kei Yip, The Chinese University of Hong Kong

The EAGER study is timely and has the potential to be a successful platform for facilitating recruitment for future trials and research studies. I have a few minor comments that I hope will enhance the clarity of this manuscript:

Reviewer 2 Comment 1

1. ****Aims and Objectives****: The primary aim is clearly articulated and pertinent to the field of autism and rare genetic conditions. However, the methodology for achieving the secondary aim is unclear, particularly since participants in EAGER are either autistic or have rare genetic conditions related to autism. Without a proper control group, it will be challenging to study the association between mental and physical health traits and participants' genetic profiles. While studying the association of a continuous trait, such as the Social Responsiveness Scale (SRS), with the genetic profile may be feasible, this hinges on whether the ascertained sample provides sufficient heterogeneity. Additionally, the manuscript does not clarify whether the sample size possesses adequate statistical power to support such an association study, nor does it address the quota allocation for ASD and non-ASD cases (but with related rare genetic conditions).

We thank the reviewer for these helpful comments. The lack of a control group is in line with recent calls to move away from case-control analyses within autism research and towards research that models differences at the level of individual participants (see Heraty et al., 2023, Lombardo et al., 2019). Within our cohort, we will be able to analyse associations between mental and physical health traits and participants' genetic profiles within autism, hence contributing to disentangling heterogeneity. Heterogeneity will be a key feature of the EAGER sample, given the diversity in participant types (e.g., wide age range, wide geographical distribution, autistic participants from simplex and multiplex families and with/without known genetic variants, participants with a range of rare genetic conditions). Analyses done in the EAGER sample (by itself, as well as integrating data from other samples such as LEAP, POND, MSSNG, SPARK, Searchlight) will help us identify as-yet-undefined genetic subgroups in autism, which can improve treatment development by linking genetic mechanisms to treatment outcomes.

In response to this comment, we have expanded the section "Data analysis" to give more detailed examples of potential analyses that can be done using EAGER data, as well as expand on the expected heterogeneity of the sample. We emphasise that these are just examples, as research questions will be decided in collaboration with community working groups. Recruitment targets were also further clarified (see response to comment 2).

"Research questions will be decided in collaboration with our community working groups, but a few brief examples are outlined below.

Using the genetic and phenotypic data, we can answer questions such as:

- Identifying genetic variants that are associated with an autism diagnosis and/or autism phenotypic characteristics (e.g., scores on SRS and SCQ).
- Identifying genetic variants that are associated with co-occurring conditions such as neurodevelopmental (e.g., scores on CAARS and ADHD rating scale) or mental health conditions (e.g., scores on SDQ, PHQ-9, and GAD-7).
- Identifying genetic variants that are associated with characteristics such as quality of life (e.g., Cantrill ladder, WHO-QOL-BREF) or functioning (e.g., general functioning questionnaire)

Using the phenotypic data, we can answer questions such as:

- Identifying participants' priorities for future research (e.g., Voice of the Community survey) and assessing the factors that may influence responses (e.g., diagnoses, age).
- Identifying associations between participant diagnoses (including co-occurring conditions) and subjective experiences such as quality of life (e.g., Cantrill ladder, WHO-QOL-BREF) or mental wellbeing (e.g., WEMWBS).

"The sample size of this study (n=1,500), which was driven by the available budget, is moderate-to-large in the field and will be significantly enhanced via the planned data sharing and integration. While EAGER will not recruit a control group, we expect there to be significant opportunities for analyses where participant subgroups can be identified and compared (e.g., participants with epilepsy compared to participants without epilepsy). The deep phenotyping and the expected heterogeneity within the sample will enable us to better understand differences between autistic people with and without a known genetic cause."

With regards to the sample size, similar research questions have been answered in much smaller samples. For example, previous studies have reported associations between genetic variants and autism and ADHD phenotype (e.g., n=50, van Daalen et al., 2011, n=688, Jansen et al., 2020, n=100, Callaghan et al., 2019) as well as quality of life and functioning (e.g., n=391, Bolbocean et al., 2021, n=176, Torske et al., 2019). Phenotypic studies have assessed associations between participant diagnoses and subjective experiences such as quality of life (e.g., n=615, Mahjob et al., 2024, n=370, Mason et al., 2018), as well as autistic participants' priorities for future research (e.g., n=130, Benevides et al., 2020). The EAGER sample will benefit from state-of-the-art genetic analysis

(i.e., whole-genome sequencing), deep phenotyping, and opportunities for joint data analysis through data sharing initiatives.

Benevides, T. W., Shore, S. M., Palmer, K., Duncan, P., Plank, A., Andresen, M. L., ... & Coughlin, S. S. (2020). Listening to the autistic voice: Mental health priorities to guide research and practice in autism from a stakeholder-driven project. *Autism, 24*(4), 822-833.

Bolbocean, C., Andújar, F. N., McCormack, M., Suter, B., & Holder Jr, J. L. (2022). Health-related quality of life in pediatric patients with syndromic autism and their caregivers. *Journal of Autism and Developmental Disorders, 52*(3), 1334-1345.

Callaghan, D. B., Rogic, S., Tan, P. P. C., Calli, K., Qiao, Y., Baldwin, R., ... & Lewis, M. S. (2019). Whole genome sequencing and variant discovery in the ASPIRE autism spectrum disorder cohort. *Clinical genetics, 96*(3), 199-206.

Heraty, S., Lautarescu, A., Belton, D., Boyle, A., Cirrincione, P., Doherty, M., ... & Jones, E. J. (2023). Bridge-building between communities: Imagining the future of biomedical autism research. *Cell, 186*(18), 3747-3752.

Jansen, A. G., Dieleman, G. C., Jansen, P. R., Verhulst, F. C., Posthuma, D., & Polderman, T. J. (2020). Psychiatric polygenic risk scores as predictor for attention deficit/hyperactivity disorder and autism spectrum disorder in a clinical child and adolescent sample. *Behavior Genetics, 50*, 203-212.

Lombardo, M. V., Lai, M. C., & Baron-Cohen, S. (2019). Big data approaches to decomposing heterogeneity across the autism spectrum. *Molecular psychiatry, 24*(10), 1435-1450.

Mahjoob, M., Cardy, R., Penner, M., Anagnostou, E., Andrade, B. F., Crosbie, J., ... & Kushki, A. (2024). Predictors of health-related quality of life for children with neurodevelopmental conditions. *Scientific Reports, 14*(1), 6377

Mason, D., McConachie, H., Garland, D., Petrou, A., Rodgers, J., & Parr, J. R. (2018). Predictors of quality of life for autistic adults. *Autism Research, 11*(8), 1138-1147.

Torske, T., Nærland, T., Bettella, F., Bjella, T., Malt, E., Høyland, A. L., ... & Andreassen, O. A. (2020). Autism spectrum disorder polygenic scores are associated with every day executive function in children admitted for clinical assessment. *Autism Research, 13*(2), 207-220.

van Daalen, E., Kemner, C., Verbeek, N. E., van der Zwaag, B., Dijkhuizen, T., Rump, P., ... & Poot, M. (2011). Social Responsiveness Scale-aided analysis of the clinical impact of copy number variations in autism. *Neurogenetics, 12*, 315-323.

Reviewer 2 Comment 2

2. ****Methodology****: Additional details regarding recruitment strategies, data collection methods, and plans for data analysis would be beneficial. For instance, the rationale behind the target sample size of N=1,500 is not specified. Is this number a result of budget constraints, subject recruitment capabilities, or a power calculation for the secondary aim? How will the 1,500 cases be allocated across different sites? Is there a quota sampling strategy in place (e.g., 50% ASD, 50% rare-genes)? Furthermore, the manuscript lacks a description of the genetic data sequencing process, such as the standardized platform and arrays used. Highlighting EAGER's standardized procedures, which I believe is one of its strengths, is recommended.

In response to this comment, we have added additional information to the manuscript with regards to the recruitment strategies, data collection methods, and plans for data analysis (see manuscript). We have also added a description of the genetic data sequencing process and have highlighted the standardised procedures (see manuscript).

In addition, we have highlighted EAGER's standardised procedures, as advised by the reviewer.

“Recruitment for EAGER will take place at 13 sites across 8 European countries (see figure 1), with the recruitment and data collection protocol being standardised across all sites.”

With regards to the sample size of 1,500, the reasoning is specified in the manuscript as
“The sample size of this study (n=1,500), which was driven by the available budget, is moderate-to-large in the field and will be significantly enhanced via the planned data sharing and integration.”

The EAGER protocol was designed to prioritise alignment of collected data with other international autism datasets which also include whole-genome sequencing and overlapping phenotypic measures such as:

- LEAP (e.g., diagnoses, SRS, RBS-R, SDQ)
- MSSNG (e.g., diagnoses, SCQ, SRS, RBS-R)
- POND (e.g., diagnoses, SCQ, RBS-R)
- Simon’s SPARK & Searchlight (e.g., Voice of the Community, diagnoses, SRS, SCQ)

We are currently piloting the joint federated analysis of data from LEAP, MSSNG, and POND through the ASI platform, with the view of using this expertise to run federated analyses using the EAGER sample, thus further maximising the power of planned analyses.

We have also added a clarification with regards to the allocation of 1,500 cases, as below:

“Recruitment targets have been allocated to each of the sites based on local capacity and cohort availability, with targets ranging from n=50 to n=120 per site. The distribution of participant types between sites also differs based on capacity, with some sites recruiting a majority of autistic participants (e.g., 90% of target sample) and other sites recruiting a majority of participants with rare genetic conditions (e.g., 70% of target sample)”

Reviewer 2 Comment 3

3. ****Ethical Considerations****: The ethical considerations, including informed consent, confidentiality, data protection, and participant privacy safeguards, are thoroughly addressed, demonstrating strong stakeholder engagement.

We thank the reviewer for this comment.

Reviewer 2 Comment 4

4. ****Data Management****: It is unclear whether ASI will serve as the host for the federated system of EAGER. The AIMS-2 Trial's website mentions the collection of data and the creation of a database that is accessible, practical, and easy to navigate. Does this refer to EAGER? Will the AIMS-2 trial data be managed by ASI (DNAstack)? Since the AIMS-2 trial was initiated a few years ago, utilising the already collected data for the development of EAGER could be more cost-effective.

We agree with the reviewer that the clarity surrounding data sharing and ASI can be improved. With this in mind, we have gone through the manuscript with the view of clarifying these points. We also directly address the reviewer’s questions here.

With regards to “It is unclear whether ASI will serve as the host for the federated system of EAGER “, indeed, the federated data analysis aspect of EAGER will happen through the ASI platform. However, data sharing will also occur through other data sharing processes as per other AIMS-2-TRIALS projects (e.g., via ELIXIR). We have now clarified this in text.

With regards to “The AIMS-2 Trial's website mentions the collection of data and the creation of a database that is accessible, practical, and easy to navigate. Does this refer to EAGER?”, we are grateful for the opportunity to clarify. From our understanding, the reviewer got this information from the Data Analysis section of the AIMS-2-TRIALS website (<https://www.aims-2-trials.eu/our-research/data-management-analysis/>).

The AIMS-2-TRIALS consortium includes a large number of studies/cohorts (e.g., Longitudinal European Autism Project, Preschool Imaging Project), with EAGER being one of them. Data from AIMS-2-TRIALS studies are stored centrally through REDcap, on the OWEY platform developed by and hosted at Institut Pasteur in Paris, France. This enables researchers within the consortium to access and use AIMS-2-TRIALS data. With regards to external data sharing, this is done in line with

the consortium data sharing policy, through mechanisms that include ELIXIR and the ASI. In response to this comment, we have now clarified this in text.

With regards to “Will the AIMS-2 trial data be managed by ASI (DNASTACK)?” the answer is no. AIMS-2-TRIALS researchers will remain data custodians and handle data access requests through established protocols. The ASI is simply one of the platforms through which data will be shared. We have now clarified this in text.

With regards to “Since the AIMS-2 trial was initiated a few years ago, utilising the already collected data for the development of EAGER could be more cost-effective.”, we understand the point being made by the reviewer. The aim of EAGER was to establish a new cohort of whole-genome sequenced and phenotyped participants who are keen to take part in future research studies and clinical trials and thus consent to being added to the registry with the view of being recontacted in the future. One of the consent items does ask for permission to use data that has already been collected by local sites, which is in line with the reviewer’s point about cost-effectiveness when using already collected data. While some of the other AIMS-2-TRIALS studies also involved whole-genome sequencing, the focus was primarily on autistic participants, with only modest-sized cohorts of rare genetic conditions (e.g., SynaG). Longer-term sustainability plans do include the potential of approaching some of the AIMS-2-TRIALS participants who are part of other cohorts to include in EAGER.

Key examples of clarifications added to the manuscript made in response to this comment are included below:

“EAGER data will also be shared via established AIMS-2-TRIALS data sharing mechanisms via platforms such as ELIXIR (<https://elixir-europe.org/>) and the Autism Sharing Initiative (ASI). The ASI is a global collaboration connecting autism data through federated mechanisms (30).”

“For this reason, EAGER data will be shared with the autism research community. Data from AIMS-2-TRIALS studies (including EAGER) are stored centrally on the OWEY platform developed by and hosted at Institut Pasteur in Paris, France. This enables researchers within the AIMS-2-TRIALS consortium to access and use the data. External data sharing is done via the ELIXIR platform, in accordance with the AIMS-2-TRIALS data sharing policy. In addition, EAGER data will be shared via the ASI platform, as detailed below.”

Reviewer 2 Comment 5

5. **Feasibility and Sustainability**: The EAGER study is funded by grant agreement No. 777394 for the AIMS-2 trials project. Does this imply that EAGER is the database referenced on the AIMS-2 trial website? If so, leveraging the existing AIMS-2 trial data for the establishment of EAGER would seem prudent. If that is the plan, then additional details about AIMS-2, particularly its timeline, are necessary. Another concern pertains to sustainability: is there sufficient funding or a realistic plan in place to ensure the registry’s longevity (e.g., management fees to ASI)?

We have clarified the reviewer’s question with regards to the database referenced on the AIMS-2-TRIALS website and leveraging existing data in the response to comment 4 above. We have added a clarification in text with regards to the timeline of AIMS-2-TRIALS and the sustainability of the project.

“AIMS-2-TRIALS is funded until May 2025. Sustainability is a core aspect of the work done as part of the consortium and plans are in place to ensure that EAGER will continue to be supported through future funding initiatives currently being pursued by the investigators.”

Reviewer 2 Comment 6

One minor comments: In Figure 1. What does R, L, S mean?

We would like to draw the attention of the reviewer to the text that is placed under the map image in Figure 1. This text specifies: R=Recruitment, L=Lead centre, S=Sample storage, sequencing, analysis. In response to this comment, we have added an asterisk (*) next to “Role” and have increased the font size for the abbreviation description.

Additional changes

In addition, we have added several authors in the author list, which were missed in the original submission, and have updated affiliations. We have updated the information in ScholarOne accordingly.